# Characteristics, Outcomes, and Factors Affecting Mortality in Hospitalized Patients with CAP Due to Different Variants of SARS-CoV-2 and Non-COVID-19 CAP

**DOI:** 10.3390/jcm12041388

**Published:** 2023-02-09

**Authors:** Nonthanat Tongsengkee, Smonrapat Surasombatpattana, Wiwatana Tanomkiat, Pisud Siripaitoon, Narongdet Kositpantawong, Siripen Kanchanasuwan, Asma Navasakulpong, Nawamin Pinpathomrat, Arunee Dechaphunkul, Atthaphong Phongphithakchai, Thanaporn Hortiwakul, Boonsri Charoenmak, Sarunyou Chusri

**Affiliations:** 1Department of Internal Medicine, Faculty of Medicine, Prince of Songkla University, Songkhla 90110, Thailand; 2Department of Pathology, Division of Virology and Serology, Faculty of Medicine, Prince of Songkla University, Songkhla 90110, Thailand; 3Department of Radiology, Faculty of Medicine, Prince of Songkla University, Songkhla 90110, Thailand; 4Department of Biomedical Sciences and Biomedical Engineering, Faculty of Medicine, Prince of Songkla University, Songkhla 90110, Thailand

**Keywords:** SARS-CoV-2, COVID-19, community-acquired pneumonia, Thailand, epidemiology, outcomes

## Abstract

The study was conducted from October 2020 to March 2022 in a province in southern Thailand. The inpatients with community-acquired pneumonia (CAP) and more than 18 years old were enrolled. Of the 1511 inpatients with CAP, COVID-19 was the leading cause, accounting for 27%. Among the patients with COVID-19 CAP, mortalities, mechanical ventilators, ICU admissions, ICU stay, and hospital costs were significantly higher than of those with non-COVID-19 CAP. Household and workplace contact with COVID-19, co-morbidities, lymphocytopenia and peripheral infiltration in chest imaging were associated with CAP due to COVID-19. The delta variant yielded the most unfavorable clinical and non-clinical outcomes. While COVID-19 CAP due to B.1.113, Alpha and Omicron variants had relatively similar outcomes. Among those with CAP, COVID-19 infection as well as obesity, a higher Charlson comorbidity index (CCI) and APACHE II score were associated with in-hospital mortality. Among those with COVID-19 CAP, obesity, infection due to the Delta variant, a higher CCI and higher APACHE II score were associated with in-hospital mortality. COVID-19 had a great impact on the epidemiology and outcomes of CAP.

## 1. Introduction

Community-acquired pneumonia (CAP) is the leading cause of hospitalization and potentially life-threatening conditions worldwide [1,2]. CAP also yields relatively high medical costs for diagnosis and management [3,4,5]. Approximately 20% of patients with CAP are hospitalized due to the severity of the disease as well as their underlying conditions [6,7,8]. Most epidemiological studies of CAP based on patients who were hospitalized, address the high fatality rate with a reported 30-day mortality rate of 13.0%, a 6-month mortality rate of 23.4%, and a 1-year mortality rate of 30.6% [7]. Various pathogens associated with CAP cause a wide range of clinical characteristics and outcomes [9,10,11]. Thus, CAP cases, due to various causative pathogens, require specific treatments and different resource utilization [12,13]. Over the decades, the epidemiology of CAP was disrupted by the emergence of infectious diseases such as the influenza pandemic, severe respiratory distress syndrome (SARS), and Middle-East Respiratory Syndrome (MERS) [14,15,16]. However, the epidemiological change occurred within a short period [14].

In December 2019, cases of CAP were reported in Wuhan, China. The etiology of these infections was severe acute respiratory syndrome coronavirus 2 (SARS-CoV-2), a novel coronavirus [17]. The infection caused due to this virus resulted in a wide range of clinical manifestations from mild symptoms, such as fever, cough, myalgia, and diarrhea, to severe forms of pneumonia and respiratory disease syndrome [18]. The SARS-CoV-2 pandemic began in early 2020 with approximately 500 million confirmed cases and 6 million deaths through August 2022 [19]. The unprecedented situation has challenged the healthcare system with a large number of hospitalizations due to CAP because of the unfamiliarity of the disease [20]. The two major concerns regarding hospitalized patients with CAP due to SARS-CoV-2 are how to distinguish them from CAP due to other causes and how to stratify them based on the different risks of unfavorable outcomes [21]. Several factors including the underlying diseases, vaccination, and treatments have been explored for their potential association with the patients’ mortality [22,23,24]. However, due to multiple epidemic waves because of different variants of this virus, the relationship between the patient outcomes and the variants remains unclear. Hence, this observational study aimed to compare the clinical and non-clinical outcomes of hospitalized patients with CAP due to SARS-CoV-2 variants and those with CAP due to other causes. 

## 2. Methodology 

### 2.1. Study Design and Population

This prospective observational study was conducted during the SARS-CoV-2 outbreaks in Thailand, initially from 1 October 2020 to 30 September 2021, then extended to 31 March 2022 (a total of 1 year and 6 months) and included patients from Songkhla Province, southern Thailand, who were admitted to the following 21 hospitals due to pneumonia: one university hospital (Songklanagarind Hospital, PSU), two provincial central hospitals (Hatyai Hospital and Songkhla Hospital), three private hospitals (Bangkok Hospital Hat Yai, Rajyindee Hospital, and Sikarin Hat Yai Hospital), and 15 district primary care hospitals (Krasae Sin Hospital, Khlong Hoi Khong Hospital, Khuanniang Hospital, Chana Hospital, Thepha Hospital, Na Mom Hospital, Bang Klam Hospital, Padang Besar Hospital, Ranot Hospital, Rattaphum Hospital, Satingpra Hospital, Somdet Phraborom Rachineenat Hospital, Sadao Hospital, Saba Yoi Hospital, and Singha Nakhon Hospital).

### 2.2. Inclusion and Exclusion Criteria 

The inclusion criteria of the patients included: (1) aged ≥18 years (2) admitted to the hospital from 1 October 2020 to 31 March 2022, and (3) diagnosed with CAP or healthcare-associated pneumonia. The exclusion criteria included patients with (1) <50% completeness of the data record, (2) an initial diagnosis of hospital-acquired pneumonia or ventilator-associated pneumonia and (3) SARS-CoV-2 co-infection with other pathogens.

### 2.3. Definition and Diagnosis Criteria

Pneumonia was diagnosed according to the modified criteria from the CDC/NHSN surveillance definition of healthcare-associated infection [25] (see Appendix A), which included clinical and imaging criteria. CAP/healthcare-associated pneumonia, (HCAP)/hospital-acquired pneumonia and (HAP)/ventilator-associated pneumonia (VAP) was diagnosed according to the ATS/IDSA definition [26] (see Appendix A).

### 2.4. Sample Collection and Pathogen Identification 

The research protocol (REC.63-164-14-1) was approved by the local ethics committee and all participants provided written informed consent before enrollment in the study. 

The doctors in charge or medical staff in each of the participating hospitals were informed to notify the research team or the correspondent, within 72 h of any patient admission due to pneumonia. Written informed consent was provided prior to enrollment. The clinical and imaging findings of the consented patients were reviewed by the research team along with a radiologist if participants are diagnosed with CAP/HCAP according to the criteria. Expectorated/endotracheal-aspirated (intubated participants) sputum was collected within 24 h of enrollment. Nasopharyngeal throat swabs were used when sputum was not adequately collected. The clinical data of the patients were collected, and the patients were followed up until they were discharged or died. Sample collection, processing, and laboratory diagnostic testing followed the World Health Organization recommendations and CDC guidelines. The patient’s sputum was re-suspended in N-acetylcysteine (NAC) in a 1:1 ratio, and the nucleic acids were extracted from 200 µL of the samples using MagDEA^®^ Dx reagents (Precision System Science, Chiba, Japan) and a fully automated nucleic acid extraction system, according to the manufacturer’s instructions. The presence of SARS-CoV-2 was detected via real-time polymerase chain reaction (RT-PCR) amplification using a SARS-CoV-2 Nucleic Acid Diagnostic Kit (Sansure, Changsha, China). ORF 1ab and N genes were used as the target regions, and human RNase P was used as an internal standard gene control with a lower limit of detection of 200 copies/mL. The respiratory pathogens were detected using the Allplex™ Respiratory Panel Assays (Seegene Inc., Seoul, South Korea), which is a multiplex one-step real-time PCR assay based on Seegene’s proprietary MuDT™ technology to identify 26 causative pathogens, including influenza virus (FluA, Flu A-H1, Flu A-H1pdm09, Flu A-H3, Flu B), respiratory syncytial virus (RSV-A, RSV-B), adenovirus, enterovirus, metapneumovirus, parainfluenza virus (PIV 1-4), bocavirus 1/2/3/4, coronavirus 229E, coronavirus NL63, coronavirus OC43, human rhinovirus, *Bordetella parapertussis*, *Bordetella pertussis*, *Chlamydophila pneumoniae*, *Hemophilus influenzae*, *Legionella pneumophila*, *Mycoplasma pneumoniae*, and *Streptococcus pneumoniae*. 

### 2.5. Statistical Analyses 

The prevalence of SARS-CoV-2 and other pathogens was reported as the proportion of participants. The continuous normally distributed data were reported as mean ± SD, the continuous skewed distributed data as median with IQRs, and the categorical data as percentages. Univariate analysis was used to compare the characteristics or clinical parameters between the two groups, and factors that showed significance (*p* value < 0.2) in the univariate analysis were then introduced into the multivariate analysis. Survival analysis was used to compare the survival among each CAP group. The statistical significance was set at a *p* value < 0.05. All statistical tests were performed using R language and environment (version 2.14.1) (Hatyai, Songkhla, Thailand).

### 2.6. Ethical Statement

This study was approved by the institutional review board of the Faculty of Medicine, Prince of Songkla University, Thailand (REC: 63-164-14-1). The researchers were given permission to retrieve clinical and microbiological data from the hospital database with a consent waiver. Before being assessed and used, all data were anonymized. The researchers confirm that this research was conducted in line with the Declaration of Helsinki principles.

## 3. Results

A total of 1526 patients were admitted to the participating hospitals in Songkhla Province due to CAP, from October 2020 to March 2022. The causative pathogens of CAP in the inpatients are shown in Table 1. SARS-CoV-2, which causes coronavirus disease 2019 (COVID-19), was the most common cause of CAP (*n* = 408), accounting for 27.0% of all the cases. The characteristics of patients with CAP due to SARS-CoV-2, its variants and those due to other causes (including unknown causes) are shown and compared in Table 2. The factors that showed a *p* value of < 0.2 were introduced into a multiple logistic regression model to differentiate between the groups (Table 3). Household or workplace contact with COVID-19 cases (*p* < 0.001), the presence of comorbidities such as diabetes mellitus (*p* = 0.002), chronic kidney disease (CKD) (*p* = 0.028), and malignancy (*p* < 0.001) were associated with the increased risk of CAP due to COVID-19. Of the 85 patients with malignancy and CAP due to COVID-19, 39 (45%) had lung cancer, 28 (33%) had hematologic malignancy, and 13 (15%) had cancer of the hepatobiliary tract. Laboratory characteristics, including lymphocytopenia (*p* < 0.001) and peripheral infiltration (*p* < 0.001) in the chest radiographic findings, were associated with CAP due to COVID-19. 

COVID-19 vaccination negatively correlated with the risk of CAP due to COVID-19 (odds ratio (OR), 0.67; 95% confidential interval (CI), 0.51–0.88). Of the patients with CAP due to COVID-19, 69 (17%) were vaccinated against SARS-CoV-2 as follows: one dose of CoronaVac (*n* = 11), two doses of CoronaVac (*n* = 7), one dose of ChAdOx1 nCoV-19 (*n* = 5), two doses of ChAdOx1 nCoV-19 (*n* = 5), one dose of BNT162b2 mRNA (*n* = 6), two doses of BNT162b2 mRNA (*n* = 6), two doses of mRNA-Moderna (*n* = 2), one dose of CoronaVac/one dose of ChAdOx1 nCoV-19 (*n* = 5), one dose of CoronaVac/one dose of BNT162b2 mRNA (*n* = 4), two doses of CoronaVac/one dose of ChAdOx1 nCoV-19 (*n* = 3), two doses of CoronaVac/one dose of BNT162b2 mRNA (*n* = 3), two doses of CoronaVac/two doses of mRNA-Moderna (*n* = 3), one dose of ChAdOx1 nCoV-19/one dose of BNT162b2 mRNA (*n* = 3), two doses of ChAdOx1 nCoV-19/one dose of BNT162b2 mRNA (*n* = 2), one dose of ChAdOx1 nCoV-19/two doses of BNT162b2 mRNA (*n* = 2), two doses of CoronaVac/two doses of ChAdOx1 nCoV-19 (*n* = 1), two doses of CoronaVac/two doses of BNT162b2 mRNA (*n* = 1).

The outcomes of patients with CAP due to COVID-19 and those with CAP due to other causes are shown and compared in Table 4. The outcomes, including mortality, length of hospital stay, and hospital costs were significantly unfavorable in patients with CAP due to COVID-19. Table 4 presents the comparison of the outcomes of patients with infection due to different strains of SARS-CoV-2. The mortality rates, lengths of hospital stay, and hospital costs significantly differed among patients infected with various strains: B. 1.113, B. 1.1.7, B. 1.617.1/2, and B. 1.1.529. The mortality rate, length of hospital stay, and hospital costs were most unfavorable in patients infected with the B.1.617.1/2 strain than those infected with other strains. Survival analysis among patients with CAP due to other causes (non-COVID-19) and CAP due to the various variants of SARS-CoV-2 demonstrated significantly different survival outcomes (*p* < 0.001 by log-rank test) (Figure 1). 

Of the 408 patients with CAP due to COVID-19, 176 died during admission, accounting for 43% of the cases. The causes of death of these patients are shown in Table 5. The factors influencing mortality in patients with CAP (*n* = 1526) included obesity, Charlson comorbidity index, initial APACHE II score, and infection due to SARS-CoV-2 (Table 6). Among the patients with CAP due to COVID-19, receiving COVID-19 vaccination and infection due to the B.1.617.1/2 strain were significantly associated with in-hospital mortality (Table 7). In patients with CAP due to COVID-19 who died during admission (*n* = 176), 20 (11%) had received COVID-19 vaccines as follows: one dose of CoronaVac (*n* = 4), two doses of CoronaVac (*n* = 3), one dose of ChAdOx1 nCoV-19 (*n* = 3), two doses of ChAdOx1 nCoV-19 (*n* = 1), one dose of BNT162b2 mRNA (*n* = 2), two doses of BNT162b2 mRNA (*n* = 1), two doses of CoronaVac/one dose of ChAdOx1 nCoV-19 (*n* = 1), two doses of CoronaVac/one dose of BNT162b2 mRNA (*n* = 1), two doses of CoronaVac/two doses of mRNA-Moderna (*n* = 1), one dose of ChAdOx1 nCoV-19/one dose of BNT162b2 mRNA (*n* = 1), two doses of ChAdOx1 nCoV-19/one dose of BNT162b2 mRNA (*n* = 1), one dose of ChAdOx1 nCoV-19/two doses of BNT162b2 mRNA (*n* = 1). 

## 4. Discussion 

This study demonstrated that among inpatients with CAP admitted to the participating hospitals between October 2020 and March 2022, SARS-CoV-2 was the most common cause of infection. The factors associated with CAP due to SARS-CoV-2 included household or workplace contact with COVID-19, and the presence of comorbidities such as diabetes mellitus, CKD, and malignancy. Clinical manifestations, including lymphocytopenia and abnormal chest radiographic findings with peripheral infiltration, were also shown to be associated with CAP due to COVID-19. Comorbidities, such as malignancy and CKD, were predominant among those with infection due to the B.1.1.529 variant. The clinical outcomes and economic burdens among patients with CAP due to COVID-19 were significantly unfavorable compared to those with CAP due to other causes. The patients with CAP due to COVID-19 infected with the B.1.617.1/2 variant yielded the most unfavorable clinical and non-clinical outcomes. The most common cause of death among inpatients with CAP due to COVID-19 was ventilator-associated pneumonia; thus, the most common causative organism was *Acinetobacter baumannii*. The factors associated with in-hospital mortality among those with CAP were obesity, Charlson comorbidity index, initial APACHE II score, and infection due to COVID-19, and among those with CAP due to COVID-19 were obesity, Charlson comorbidity index, initial APACHE II score, and infection due to the B.1.617.1/2 variant. 

Regarding the inclusion criteria, the current study focused only on inpatients and did not include those with mild symptoms, such as walking pneumonia or patients with CAP due to COVID-19 admitted in the community or home isolation. Therefore, several patient characteristics should be considered. First, the clinical condition of the enrolled patients was relatively severe with an initial APACHE score of 17 and a respiratory failure rate of 11% at admission, when compared to previous data on the patients with CAP [27]. Second, the percentage of identifiable causative pathogens were higher (75.2%) in our study cohort than in previous reports [9,10,11,28,29,30]. These can be explained by the obvious presenting symptoms of the patients; for example, a large amount of sputum or intubation for mechanical ventilation aiding specimen (sputum) collection and raised the sensitivity for diagnosis. Thus, the current study used an additional molecular method with a microarray technique in addition to conventional culture methods for the identification of the causative pathogens [31,32] Additionally, the major causative pathogen was SARS-CoV-2, identified using a molecular technique of PCR from sputum and nasal/nasopharyngeal swabs. Third, the proportion of pathogens causing non-severe or walking pneumonia, such as *Mycoplasma* spp. was comparatively low [33,34,35].

Several clinical characteristics of patients with CAP due to SARS-CoV-2 have been demonstrated and distinguished from those with CAP due to other pathogens. The comorbidities, including diabetes mellitus, CKD, and malignancy were significantly associated with CAP due to SARS-CoV-2 among those with CAP. In this study, the presence of diabetes mellitus was found to be significantly associated with CAP due to SARS-CoV-2. Although the prevalence of diabetes among COVID-19 patients varied from 7.7 to 35% (higher in the US compared with China) and is not higher than that in the general population, diabetes or uncontrolled hyperglycemia tends to have an unfavorable outcome [33] (e.g., severe symptoms, acute respiratory distress syndrome (ARDS), ICU admission, and mortality). Several mechanisms have been proposed to explain this phenomenon including hyperglycemia-induced lung dysfunction, immunosuppression, and increased oxidative stress (OS). Diabetes-induced OS plays a major role in COVID-19 pneumonia progression. Hyperglycemia promotes OS by elevating mitochondrial superoxide anion generation and increasing the glycosylation of proteins, as well as by activating various signaling pathways that may change pulmonary function and structure (e.g., decrease in volume, elastic recoil, and diffusion capacity) [34].

The present study also showed that patients with underlying malignancy had significantly more complications with CAP due to COVID-19 than those with non-COVID-19 CAP. Our results are concordant with those of previous studies demonstrating an increased risks of COVID-19-related severe events and mortality rates in patients with cancer [36,37,38,39]. Furthermore, the independent factors reported to be associated with increased 30-day mortality in cancer patients with COVID-19 included older age, male sex, former smoking habit, presence of comorbidities, an Eastern Cooperative Oncology Group performance status of 2 or higher, and receipt of azithromycin plus hydroxychloroquine [36]. A recent publication from India also demonstrated that advancing age, smoking history, concurrent comorbidities, and palliative intent of treatment were independently associated with severe COVID-19 or death [39]. In addition, patients with cancer might have a higher risk of COVID-19 than those without cancer in Chinese populations [38]. This finding might be potentially explained by the fact that the expression of angiotensin-converting enzyme 2 (ACE2) increases with age, and, in general, cancer patients tend to be older. Since ACE2 is a receptor for the virus permitting entry into the target cells, a higher expression of ACE2 would escalate the risk of SARS-CoV-2 infection in these patients [40]. Taken together, whether cancer itself or the aforementioned risks commonly observed in cancer patients increase the risk of COVID-19 remains unclear. This study also found that patients with lung cancer appeared to have a greater risk of CAP due to COVID-19 than those with other types of malignancies. Interestingly, Gottschalk et al. reported that ACE2, which is frequently detected in patients with lung cancer, is also strongly up-regulated in the lungs during SARS-CoV-2 infection. Therefore, the up-regulated expression of ACE2 in lung tumors might increase the susceptibility to SARS-CoV-2 infection in patients with lung cancer [41]. In contrast, another study reported no difference in severe events related to COVID-19 between patients with lung cancer and those with other cancers [38]. However, it is important to note that the number of patients with lung cancer in that study was quite small (*n* = 5); hence, it was unlikely to detect significant differences between the groups. Additionally, patients with lung cancer might be at an increased risk for SARS-CoV-2 infection due to difficulties in prolonged mask-wearing due to their compromised respiratory status [42].

Our study showed that CKD was associated with an increased risk of CAP due to SARS-CoV-2, which was corroborated by the large national wide analysis from China which reported that patients with COVID-19 and CKD had a high possibility of ICU admission and receiving invasive mechanical ventilation and thus a higher mortality rate [43]. In contrast, the studies discussed in a meta-analysis reported that CKD is not a predictor of severe SARS-CoV-2 infection. Nevertheless, after the pool estimation, the presence of CKD increased the risk of severe COVID-19 [44]. This might be due to the inclusion of underpowered studies that did not have a sufficient sample size to detect a true effect. Recently, a study from the US demonstrated that CKD is a significant predictor of mortality among patients with COVID-19 after adjusting for other risk factors [45]. To our knowledge, these associations could be explained by two main reasons: first, uremia-induced dysregulation of the immune system revealed inadequate CD4 T cell responses, which led to a delayed clearance of the virus [46,47,48]. Second, ACE2 receptor, used by SARS-CoV-2 for initial attachment and access into the cells, is highly expressed in the lungs and kidneys. Hence, this might increase a patient’s susceptibility to adverse outcomes [49,50]. Interestingly, this study found that CKD strongly correlated with CAP due to the B.1.159 SARS-CoV-2 variant, which was similar to a previous report that showed that renal disease was a common comorbidity among patients infected with SARS-CoV-2 [51]. However, they also found that patients with COVID-19 were less likely to have severe clinical consequences. Thus, we hypothesized that the effect of renal disease on CAP due to the B.1.159 variant could be related to the residual confounding by the increased number of COVID-19 CKD cases per day during the Omicron period. In fact, there was a decline in COVID-19 patient admissions during the B.1.159 SARS-CoV-2 variant outbreak. Hence, patients with CKD who were unvaccinated against SARS-CoV-2 showed severe symptoms, as demonstrated by CAP, and were more likely to seek medical care at the hospital during the Omicron wave. However, this approach can also lead to the overestimation of the number of patients.

Several investigative laboratories have distinguished patients with CAP due to SARS-CoV-2 from those with CAP due to other pathogens. Leukopenia and lymphopenia were higher in patients with COVID-19 than in non-COVID-19 pneumonia patients; however, after adjusting for OR with other parameters, only leukopenia was associated with COVID-19 pneumonia. A meta-analysis of data on 1995 patients with COVID-19 reported that 64.5% of the patients had lymphocytopenia and 29.4% had leukocytopenia [52]; however, the underlying mechanism is not well understood. Granulocytic myeloid-derived suppressor cells (G-MDSCs) have been proposed to play an important role in T lymphocyte suppression. The suppressive functions of G-MDSCs include (1) impaired T cell proliferation and suppressed IFN-γ cytokine production; (2) anergy of effector CD8+ cytotoxic T cells and CD4+ helper T cells, or (3) expansion of Treg cells [53]. This study showed that peripheral infiltration on chest radiographs was associated with CAP due to COVID-19. Multiple meta-analyses reported that the most common chest radiographic findings were ground-glass opacity and consolidation, which involve the bilateral lungs and have peripheral distribution [54,55,56], which could be due to the involvement of the SARS-CoV-2 infection site. SARS-CoV-2 is usually involved in gas exchange units, especially pneumocyte type II cells, which infect the peripheral and subpleural areas, resulting in the infiltration patterns [57].

The current study demonstrated that receiving a vaccination against COVID-19 was negatively associated with the risk of CAP due to COVID-19 and in-hospital mortality. The efficacy of COVID-19 vaccination in this study should be carefully considered owing to several limitations. First, there was relatively high heterogeneity in the types of vaccines administered with varying effectiveness in infection prevention and decreased the severity of the disease [58,59,60,61]. Second, the variation of completed vaccination (at least 2 dosages) course as well as the boosting dosage(s) among the patients receiving vaccines against COVID-19 might affect the effectiveness of each vaccination [58,59,60,61]. Third, in Thailand, there were several heterogeneous regimens for COVID-19 vaccination, which should be considered when analyzing the varying outcomes [62,63]. Finally, each variant of SARS-CoV-2 emerged in different periods with a wide range of rates of completed vaccination and boosting. Thus, recent influenza immunization provided borderline protective mortality against CAP due to COVID-19. It is well established that COVID-19 vaccination and infection reduce disease mortality by inducing antigen-specific T cell repertoires. Re-infection with different viral strains is mainly asymptotic, which could be explained by the well-conserved region of spike-specific T cells [64,65]. Influenza vaccines induce T cell responses specific to the influenza nucleocapsid, which can cross-counter the nucleocapsid of other common cold viruses, including corona species [66,67]. Therefore, pre-existing nucleocapsid-specific T cells could confer some protection against SARS-CoV-2 infection.

The present study also found that the severity and mortality of patients with COVID-19 differed among the variants. The B.1.617.1/2 (Delta) variant was a risk factor for unfavorable outcomes, whereas pre-Delta and its predecessors compared with Omicron had similar ICU admission rates and mortality, similar to previous studies that reported the association of the B.1.617.1/2 variant with poor outcomes and the Omicron variant with a better outcome [68,69,70,71,72,73]. The disease severity associated with Alpha, Gamma, and Delta variants was reported to be comparable, while Omicron infections were significantly less severe, but the breakthrough disease was significantly more common in patients with Omicron infection [74]. While in this study, the mortality among patients infected with the Omicron variant was similar to that of the B.1.113 and B.1.1.7 (Alpha) variants. Magazine et al. reported more related mutations located in the S1 subunit of the spike protein such as 69-70del, 144del, and 156del, including N501Y and D614G, between the Alpha and Omicron variants when compared with other variants, which play an essential role in enhancing viral replication and increasing transmission [75,76,77]. The 1.617.1/2 variant developed several mutations in the viral genome that resulted in many significant features [78]. The Delta spike mutation P681R, located at the furin cleavage site, enhances the cleavage of the full-length spike to S1 and S2 subunits, which improves cell-surface-mediated viral entry [79]. The L452R and T478K mutations enhance ACE receptor affinity by altering the receptor-binding domain (RBD) and developing immune invasion (both immune and vaccine escape) due to changes in the epitope of the spike protein [78]. Immune invasion results in a higher viral load and a longer viral shedding time [73]. When compared to other earlier studies, this one found a relatively low incidence of COVID-19-associated pulmonary aspergillosis (CAPA), with an estimated incidence of 8.6% in patients receiving mechanical ventilation [80]. This could be explained by the fact that COVID-19 CAP patients were admitted in a newly constructed isolated or negative pressure room instead of an old one; before the COVID-19 pandemic, hospitals had a very low number of negative pressure or isolated rooms. Therefore, the new one may have less fungal contamination and reduce the incidence of aspergillosis and mucormycosis.

Although previous studies demonstrated the efficacy of systemic corticosteroids, remdesivir, interleukin-6 inhibitors, and Janus kinase inhibitors for patients with severe CAP due to SARS-CoV-2 [81,82,83,84,85], those efficacies were not demonstrated in this current study. It can be explained as follows: (1) due to the wide range of severity of patients in this study, patients with mild disease might not receive the benefits from those treatments. (2) Most of the patients were treated according to the standard guidelines, which recommend using these treatments based on the severity of the patient.

This study has some limitations that should be acknowledged. First, the patients with CAP and mild symptoms were not enrolled in this study. Second, this study population could have included patients with healthcare-associated pneumonia, such as those undergoing regular hemodialysis or receiving chemotherapy. Third, the potential misclassification bias might be secondary to tertiary data extraction. Fourth, in relation to non-interventional studies, the reason for clinical decisions has not been well explored. Lastly, there were several interferences from the dynamics of diseases, including changes in major variants involved in outbreaks over time, changes in the national policy for the standard of treatment, and availability of those recommended treatments, which potentially might alter the clinical outcomes of infection. 

In conclusion, during the COVID-19 pandemic, SARS-CoV-2 was the major cause of CAP and caused unfavorable clinical and non-clinical outcomes, particularly due to the B.1.617.1/2 variant. Regarding to the wide range of vaccine regimens and dosages, even this study demonstrated that the COVID-19 vaccination tended to be protective factor; thus, the efficacy of vaccination should be further explored.

## Figures and Tables

**Figure 1 jcm-12-01388-f001:**
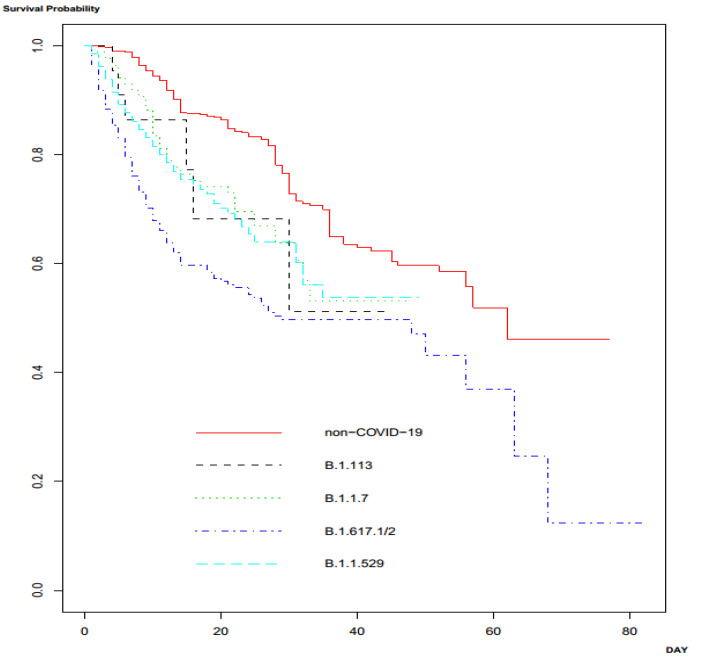
Survival analysis among the patients with community-acquired pneumonia (CAP) due to SARS-CoV-2 variants and other pathogens.

**Table 1 jcm-12-01388-t001:** Causative pathogens of community-acquired pneumonia (CAP) in the study cohort.

Causative Pathogens	Frequency (%) (*n* = 1511)
Bacterial pathogens	
*Streptococcus pneumoniae*	284 (18.8)
*Staphylococcus aureus*	59 (3.9)
*Klebsiella pneumoniae*	53 (3.5)
Other Enterobacteriaceae	31 (2.1)
*Pseudomonas aeruginosa*	13 (0.9)
*Streptococcus* spp.	9 (0.6)
*Acinetobacter baumannii*	9 (0.6)
*Burkhodelia pseudomallei*	8 (0.5)
*Hemophilus influenzae*	4 (0.3)
*Mycoplasma pneumoniae*	2 (0.1)
**Viral pathogens**	
SARS-CoV-2	408 (27.0)
Influenza A	107 (7.1)
Influenza B	61 (4.0)
Respiratory syncytial virus A	12 (0.8)
Respiratory syncytial virus B	11 (0.7)
Adenovirus	9 (0.6)
Enterovirus	9 (0.6)
Parainfluenza virus	8 (0.5)
Coronavirus 229E	7 (0.5)
Coronavirus NL63	6 (0.4)
Coronavirus OC43	6 (0.4)
Bocavirus	3 (0.2)
Metapneumovirus	3 (0.2)
Rhinovirus	1 (0.1)
Unknown cause	374 (24.8)

**Table 2 jcm-12-01388-t002:** Characteristics of patients with community-acquired pneumonia (CAP) and comparison between CAP due to coronavirus disease 2019 (COVID-19) and CAP due to non-COVID-19.

Characteristics	Patients with CAP (*n* = 1511)	Patients with COVID-19 CAP (*n* = 408)	Patients with Non-COVID-19 CAP (*n* = 1103)	*p* Value ^A^	Patients with CAP Due to B.1.113 (*n* = 22)	Patients with CAP Due to B.1.1.7 (*n* = 85)	Patients with CAP Due to B.1.617.1/2 (*n* = 171)	Patients with CAP Due to B.1.1.529 (*n* = 130)	*p* Value ^B^
Demographic data									
Age; median (IQR)	61 (50.73)	61 (51.73)	60 (49.72)	0.877	60 (52.74)	62 (55.75)	60 (50.69)	61 (52.68)	0.568
Male sex	824 (55)	229 (56)	595 (54)	0.449	11 (50)	49 (58)	94 (55)	75 (58)	0.860
Living in an urban area	663 (44)	167 (41)	496 (45)	0.161	10 (45)	33 (39)	72 (42)	52 (40)	0.417
Healthcare occupation	26 (2)	8 (2)	18 (2)	0.663	1 (5)	1 (1)	3 (2)	3 (2)	0.760
Current smokers	559 (37)	151 (37)	408 (37)	0.994	7 (32)	34 (40)	62 (36)	48 (37)	0.878
COVID-19 vaccination	364 (24)	69 (17)	295 (27)	**<0** **.001**	2 (9)	11 (13)	26 (15)	30 (23)	0.117
Recent Influenza vaccination	212 (14)	52 (13)	160 (15)	0.382	2 (9)	12 (14)	24 (14)	14 (11)	0.785
Pneumococcal vaccination	53 (4)	13 (3)	40 (4)	0.680	1 (5)	3 (4)	6 (4)	4 (3)	0.987
Household contact with COVID-19	406 (27)	293 (72)	113 (10)	**<0** **.001**	13 (59)	60 (70)	123 (72)	97 (75)	0.507
Workplace contact with COVID-19	98 (6)	78 (19)	20 (2)	**<0** **.001**	3 (14)	14 (16)	35 (20)	26 (20)	0.782
Comorbidities									
Obesity	417 (28)	117 (29)	300 (27)	0.568	5 (23)	25 (29)	50 (29)	37 (28)	0.933
Diabetes mellitus	467 (31)	156 (38)	311 (30)	**0** **.002**	8 (36)	31 (36)	66 (39)	51 (39)	0.976
Hypertension	616 (41)	163 (40)	453 (41)	0.694	9 (41)	38 (45)	65 (38)	51 (39)	0.776
Dyslipidemia	616 (41)	165 (40)	451 (41)	0.875	10 (45)	37 (44)	68 (40)	50 (38)	0.847
Chronic kidney disease	490 (32)	150 (37)	340 (31)	**0.028**	6 (27)	26 (31)	54 (32)	64 (49)	**0** **.005**
Cardiovascular disease	242 (16)	70 (17)	172 (16)	0.462	4 (18)	13 (15)	30 (18)	23 (18)	0.966
Cerebrovascular disease	148 (10)	39 (10)	109 (10)	0.851	3 (14)	9 (11)	15 (9)	12 (9)	0.881
Pulmonary disease	412 (27)	107 (26)	301 (27)	0.679	5 (23)	23 (27)	50 (29)	29 (22)	0.572
Liver disease	106 (7)	31 (8)	77 (7)	0.674	4 (18)	7 (8)	11 (6)	9 (7)	0.264
Malignancy	225 (15)	85 (20)	140 (13)	**<0** **.001**	3 (14)	13 (15)	28 (16)	41 (32)	**0** **.004**
HIV infection	44 (3)	13 (3)	31 (3)	0.700	1 (5)	3 (4)	4 (2)	5 (4)	0.862
Rheumatologic disease	63 (4)	21 (5)	41 (4)	0.216	1 (5)	4 (5)	10 (6)	6 (5)	0.981
Immunocompromised status	103 (7)	29 (7)	74 (7)	0.784	2 (9)	7 (8)	13 (8)	7 (5)	0.815
Charlson comorbidity index, median (IQR)	6 (5,8)	6 (5,8)	6 (5,9)	0.768	6 (4,7)	6 (5,8)	6 (5,7)	6 (5,8)	0.943
At least one comorbidity	1128 (75)	301 (74)	827 (75)	0.633	17 (77)	67 (79)	123 (72)	94 (72)	0.636
Clinical data									
Fever	1337 (88)	356 (87)	981 (89)	0.362	20 (91)	70 (82)	157 (92)	109 (84)	0.084
Upper respiratory tract prior to pneumonia	749 (50)	209 (51)	540 (49)	0.434	13 (59)	43 (51)	89 (52)	64 (49)	0.847
Initial respiratory failure/ Initial mechanical ventilator	164 (11)	45 (11)	119 (11)	0.984	1 (5)	6 (7)	29 (17)	9 (7)	**0** **.013**
Initial APACHE II score	17 (15,20)	17 (15,20)	17 (14,20)	0.805	16 (14,19)	16 (14,20)	18 (15,21)	15 (14,19)	0.052
Initial multiorgan failure	30 (2)	11 (3)	19 (2)	0.232	1 (5)	2 (2)	5 (3)	3 (2)	0.133
Initial ICU admission	71 (5)	34 (8)	37 (3)	**<0** **.001**	2 (1)	4 (5)	20 (12)	6 (5)	**<0** **.001**
Initial vasopressor	15 (1)	6 (1)	9 (1)	0.768	1 (5)	1 (1)	3 (2)	1 (1)	0.675
Laboratory data									
Anemia	122 (8)	32 (8)	90 (8)	0.841	2 (9)	6 (7)	15 (9)	9 (7)	0.924
Leukopenia	154 (10)	113 (28)	41 (3)	**<0** **.001**	3 (14)	18 (21)	62 (36)	30 (23)	**0** **.008**
Lymphocytopenia	107 (7)	81 (20)	26 (2)	**<0** **.001**	3 (14)	13 (15)	49 (29)	16 (12)	**0** **.002**
Thrombocytopenia	76 (5)	25 (6)	51 (5)	0.236	2 (9)	5 (6)	11 (6)	7 (5)	0.920
Abnormal liver function test	171 (11)	48 (12)	124 (11)	0.776	2 (9)	12 (14)	20 (12)	14 (11)	0.866
Abnormal renal function test	513 (34)	161 (39)	352 (32)	**0** **.006**	6 (27)	27 (32)	58 (34)	70 (54)	**<0** **.001**
Chest radiological findings									
Bilateral infiltration	859 (57)	239 (59)	620 (56)	0.409	13 (59)	52 (61)	103 (60)	71 (55)	0.735
Peripheral infiltration	322 (21)	207 (51)	115 (10)	**<0** **.001**	10 (45)	46 (54)	84 (49)	67 (52)	0.838
Multi-lobar infiltration	857 (57)	250 (61)	647 (59)	0.358	11 (50)	54 (64)	107 (63)	78 (60)	0.488
Ground glass opacity	184 (12)	51 (13)	133 (12)	0.212	4 (18)	11 (13)	25 (15)	11 (8)	0.347

^A^ Comparison between patients with Coronavirus Disease 2019 (COVID-19) CAP and patients with non-COVID-19 CAP, ^B^ Comparisons among patients with CAP due to B.1.113, B.1.1.7, B.1.617.1/2 and B.1.1.529, Boldface entries indicate values that reached the significance level set at 0.05.

**Table 3 jcm-12-01388-t003:** Characteristics associated with community-acquired pneumonia (CAP) due to coronavirus disease 2019 (COVID-19).

Selected Variables	Patients with COVID-19 CAP *n* = 408 (%)	Patients with Non-COVID-19 CAP *n* = 1103 (%)	Crude OR (95% CI)	Adjusted OR (95% CI)	*p* Value ^A^
Living in an urban area	167 (41)	496 (45)	0.85 (0.67–1.07)	0.78 (0.54–1.03)	0.231
COVID-19 vaccination	69 (17)	295 (27)	0.56 (0.42–0.75)	0.67 (0.51–0.88)	**0** **.002**
Household contact with COVID-19	293 (72)	113 (10)	22.32 (16.69–29.85)	16.52 (9.11–20.98)	**<0** **.001**
Workplace contact with COVID-19	78 (19)	20 (2)	12.80 (7.71–21.24)	9.07 (6.55–17.34)	**<0** **.001**
Diabetes mellitus	156 (38)	311 (30)	1.44 (1.14–1.83)	1.26 (1.03–1.56)	**0** **.021**
Chronic kidney disease	150 (37)	340 (31)	1.30 (1.03–1.66)	1.12 (1.01–1.42)	**0** **.042**
Malignancy	85 (20)	140 (13)	1.81 (1.34–2.44)	1.45 (1.17–2.00)	**0** **.004**
Leukopenia	113 (28)	41 (3)	9.92 (6.78–14.51))	4.08 (0.92–6.65)	0.086
Lymphocytopenia	81 (20)	26 (2)	10.26 (6.48–16.23)	5.51 (1.97–7.71)	**0** **.003**
Abnormal renal function test	161 (39)	352 (32)	1.39 (1.10–1.76)	1.07 (0.82–1.20)	0.089
Chest radiological finding with peripheral infiltration	207 (51)	115 (10)	8.85 (6.73–11.63)	4.43 (2.17–7.93)	**0** **.002**

CAP, community-acquired pneumonia; COVID-19, coronavirus disease 2019; OR, odds ratio; CI, confidence interval. A Comparison between patients with COVID-19 CAP and non-COVID-19 CAP. Boldface entries indicate values that reached the significance level set at 0.05.

**Table 4 jcm-12-01388-t004:** Comparisons of outcomes of patients with community-acquired pneumonia (CAP) due to coronavirus disease 2019 (COVID-19) and non-COVID-19 CAP and comparisons of outcomes among the patients with CAP due to the B.1.113, B.1.1.7, B.1.617.1/2 and B.1.1.529 variants.

Outcomes	Patients with COVID-19 CAP (*n* = 408)	Patients with Non-COVID-19 CAP (*n* = 1103)	*p* Value ^A^	Patients with CAP Due to B.1.113 (*n* = 22)	Patients with CAP Due to B.1.1.7 (*n* = 85)	Patients with CAP Due to B.1.617.1/2 (*n* = 171)	Patients with CAP Due to B.1.1.529 (*n* = 130)	*p* Value ^B^
Mortality								
14-day	124 (30)	109 (10)	**<0** **.001**	3 (14)	20 (24)	69 (40)	32 (25)	**<0** **.001**
30-day	163 (40)	162 (15)	**<0** **.001**	7 (32)	29 (34)	85 (50)	42 (32)	**<0** **.001**
In-hospital	176 (43)	199 (18)	**<0** **.001**	8 (32)	30 (35)	90 (53)	48 (37)	**<0** **.001**
Length of ICU stay, median (IQR)	17 (14,39)	9 (7.14)	**<0** **.001**	15 (14,16)	15 (14,20)	19 (14.40)	14 (14.17)	**0** **.031**
Length of hospital stay, median (IQR) after survival	14 (11.26)	25 (21.34)	**<0** **.001**	21 (21,25)	24 (21,33)	38 (33,45)	24 (21.38)	**0** **.026**
Hospital cost, median (IQR)	97,007 (82,876–107,664)	87,158 (64,009–97,425)	**<0** **.001**	85,662 (69,557–96,457)	99,623 (85,965–100,885)	99,881 (85,988–110,231)	98,997 (85,923–99,227)	**0** **.042**
Antimicrobials cost	35,001 (31,732–44,013)	20,884 (17,654–23,112)	**<0** **.001**	33,881 (30,445–42,985)	35,998 (32,878–45,954)	36,112 (33,454–46,009)	35,080 (33,656–46,543)	0.087
Non-antimicrobials cost	66,881 (55,054–74,992)	64,903 (54,243–73,775)	**0** **.002**	60,201 (52,881–70,320)	66,995 (57,884–76,362)	68,990 (61,332–79,881)	66,121 (58,098–77,441)	**0** **.019**

CAP, community-acquired pneumonia; COVID-19; coronavirus disease 2019; IQR, interquartile range. A Comparison between patients with COVID-19 CAP and non-COVID-19 CAP. B Comparisons among patients with CAP due to the B.1.113, B.1.1.7, B.1.617.1/2, and B.1.1.529 variants. Boldface entries indicate values that reached the significance level set at 0.05.

**Table 5 jcm-12-01388-t005:** Causes of in-hospital deaths in patients with community-acquired pneumonia (CAP) due to coronavirus disease 2019 (COVID-19).

Causative Pathogens	Number (%) (*n* = 176)
Ventilator association pneumonia due to	85 (48)
*Acinetobacter bauamnnii*	52 (30)
Carbapenem-resistant *A*. *bauamnnii*	52 (30)
*Staphylococcus aureus*	12 (7)
Methicillin-resistant *S*. *aureus*	2 (1)
*Pseudomonas aeruginosa*	7 (4)
Carbapenem-resistant *P*. *aeruginosa*	7 (4)
*Klebsiella pneumoniae*	3 (2)
Extended-spectrum beta-lactamase *K*. *pneumoniae*	1 (1)
Carbapenem-resistant *K*. *pneumoniae*	2 (1)
Unknown pathogen	11 (6)
**Bloodstream infection due to**	50 (28)
*Staphylococcus Aureus*	14 (8)
Methicillin-resistant *S*. *aureus*	3 (2)
*Acinetobacter bauamnnii*	13 (7)
Carbapenem-resistant *A*. *bauamnnii*	13 (7)
*Pseudomonas aeruginosa*	11 (6)
Carbapenem-resistant *P*. *aeruginosa*	9 (5)
*Klebsiella pneumoniae*	9 (5)
Extended-spectrum beta-lactamase *K*. *pneumoniae*	3 (2)
Carbapenem-resistant *K*. *pneumoniae*	6 (3)
*Candida* spp.	3 (2)
Invasive mold infection	9 (5)
Paranasal sinusitis due to	5 (3)
*Aspergillus* spp.	2 (1)
*Mucor* spp.	2 (1)
*Cunninghamella* spp.	1 (0.5)
Brain abscess due to	2 (1)
*Aspergillus* spp.	1 (0.5)
*Rhizopus* spp.	1 (0.5)
Pulmonary infection due to	2 (1)
*Aspergillus* spp.	1 (0.5)
*Mucor* spp.	1 (0.5)
Bleeding disorder	8 (5)
Gastrointestinal bleeding	3 (2)
Intracranial bleeding	3 (2)
Pulmonary bleeding	2 (1)
Thromboembolism	8 (5)
Pulmonary embolism	3 (2)
Myocardial infarction	3 (2)
Cerebral infarction	2 (1)
Unknown cause	16 (9)

**Table 6 jcm-12-01388-t006:** Factors associated with in-hospital mortality among the patients with community-acquired pneumonia (CAP).

Variables	Survivors *n* = 1136 (%)	Non-Survivors *n* = 375 (%)	Crude OR (95% CI)	Adjusted OR (95% CI)	*p* Value ^A^
Age (years) [median (IQR)]	63 (52.69)	64 (54.73)	1.61 (1.11, 2.08)	1.05 (0.94, 1.76)	0.076
Male sex	619 (54)	205 (55)	1.01 (0.80, 1.27)	1.00 (0.65, 1.12)	0.966
Living in an urban area	475 (42)	188 (50)	1.40 (1.11, 1.77)	1.24 (0.98,1.52)	0.063
Obesity	260 (23)	157 (42)	2.43 (1.89, 3.11)	1.98 (1.27, 2.65)	0.009
Current smoking	392 (35)	167 (45)	1.52 (1.20, 1.93)	1.28 (0.97, 1.66)	0.060
COVID-19 vaccination	298 (26)	66 (18)	0.60 (0.45, 0.81)	0.91 (0.78, 1.02)	0.059
Recent influenza vaccination	180 (16)	32 (9)	0.50 (0.33,0.74)	0.82 (0.68, 1.04)	0.058
Charlson comorbidity index, median (IQR)	4 (3,6)	7 (5,8)	1.57 (1.23, 1.88)	1.25 (1.08, 1.59)	0.039
Immunocompromised status	69 (6)	34 (9)	1.54 (1.00, 2.36)	1.19 (0.84, 1.97)	0.071
Initial respiratory failure	113 (10)	51 (14)	1.43 (0.99, 2.03)	1.18 (0.87, 1.84)	0.082
Initial APACHE II score	16 (14,17)	17 (15.18)	1.56 (1.08, 2.07)	1.19 (1.00, 1.91)	0.050
CAP due to COVID-19	232 (20)	176 (47)	3.44 (2.69, 4.42)	2.85 (1.97, 3.43)	<0.001

CAP, community-acquired pneumonia; COVID-19; coronavirus disease 2019; APACHE II, Acute Physiology and Chronic Health Evaluation II, IQR; interquartile range; OR, odds ratio; CI, confidence interval. ^A^ Comparison between survivors and non-survivors. Boldface entries indicate values that reached the significance level set at 0.05.

**Table 7 jcm-12-01388-t007:** Factors associated with in-hospital mortality among the patients with community-acquired pneumonia (CAP) due to due to coronavirus disease 2019 (COVID-19).

Variables	Survivors *n* = 232 (%)	Non-Survivors *n* = 176 (%)	Crude OR (95% CI)	Adjusted OR (95% CI)	*p* Value ^A^
Age (year) [median (IQR)]	60 (50.71)	61 (52.73)	1.09 (0.76, 1.29)	1.03 (0.87, 1.16)	0.786
Male sex	133 (56)	96 (55)	0.89 (0.60, 1.33)	0.75 (0.51, 1.10)	0.674
Living in an urban area	91 (39)	76 (43)	1.18 (0.79, 1.75)	1.05 (0.61, 1.23)	0.577
Obesity	53 (23)	64 (36)	1.93 (1.25, 2.98)	1.42 (1.04, 2.00)	**0** **.** **021**
Current smoking	78 (34)	73 (41)	1.73 (1.15, 2.60)	1.23 (0.87, 1.87)	0.081
COVID-19 vaccination	49 (21)	20 (11)	0.48 (0.27, 0.84)	0.76 (0.56, 0.91)	**0** **.** **037**
Recent influenza vaccination	37 (16)	15 (9)	0.49 (0.26, 0.92)	0.59 (0.42, 1.02)	0.052
Charlson comorbidity index, median (IQR)	5 (4.7)	7 (5.8)	1.43 (1.21, 1.76)	1.18 (1.03, 1.43)	**0** **.** **027**
Immunocompromised status	15 (6)	14 (8)	1.25 (0.59, 2.66)	1.09 (0.84, 1.98)	0.713
Initial respiratory failure	18 (8)	27 (15)	2.15 (1.15, 4.05)	1.54 (0.97, 2.84)	0.064
Initial APACHE II score	15 (13.17)	18 (14.19)	1.37 (1.15, 2.08)	1.18 (1.09, 1.64)	**0** **.** **018**
Infection due to B.1.617.1/2	83 (36)	90 (52)	1.88 (1.26, 2.80)	1.22 (1.05, 1.94)	**0** **.** **030**
Receiving favipiravir within 5 days of symptoms prior to pneumonia	125 (54)	92 (52)	0.94 (0.63–1.39)	0.96 (0.71–1.56)	0.823
Receiving remdesivir during treatment	222 (96)	169 (96)	1.09 (0.41–2.92)	1.01 (0.31–2.24)	0.991
Receiving corticosteroid during treatment	211 (91)	154 (88)	0.70 (0.37–1.31)	0.81 (0.45–1.28)	0.384
Receiving Interleukin-6 Inhibitors/Janus Kinase inhibitors during treatment	95 (41)	65 (37)	0.84 (0.56–1.26)	0.89 (0.66–1.31)	0.489

APACHE II: Acute Physiology and Chronic Health Evaluation II, IQR; interquartile range; OR, odds ratio; CI, confidence interval. ^A^ Comparison between survivors and non-survivors. Boldface entries indicate values that reached the significance level set at 0.05.

## Data Availability

Data is contained within the article.

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
