# Peer review of "Characteristics, Outcomes, and Factors Affecting Mortality in Hospitalized Patients with CAP Due to Different Variants of SARS-CoV-2 and Non-COVID-19 CAP"

_jcm, 2023, doi:10.3390/jcm12041388_

Round 1
Reviewer 1 Report
I evaluated the study.
In this study, the authors present a study called "Features, outcomes, and factors affecting mortality in hospitalized patients with community-acquired pneumonia due to different SARS-CoV-2 variants", as the title suggests.
Some important criticisms of the study, which I have mentioned in the order, are shown below.
1. From the name of the study, it is likely understood that all patients consisted of patients with pneumonia caused by COVID-19 variants. However, according to the method part, the rate of CAP associated with COVID-19 is 27%. For this reason, I think it would be more appropriate to change the title. For example; Characteristics, outcomes, and factors affecting mortality in hospitalized patients with CAP due to different variants of SARS-CoV-2 and non-COVID-19 CAP.
2. Patients excluded from the analysis should not have been mentioned or included anywhere else in the text, except for the method section. However, the authors mentioned 15 cases with co-infection in the results section.
3. As on 130. line stated, If 15 cases had been excluded, it is an erroneous approach to indicate the data of excluded patients in the results section. As I mentioned earlier, the exclusion criteria must only be in the method section. It would be wrong to show excluded cases in the results. This line needs to be corrected. The exclusion criteria should also be reviewed.
4. In Table 1, All patients (including those with co-infection)=1526-15 (Co-infection)=1511. I don't understand why the authors show the excluded cases along with all cases here in Table 1. Why the patients with co-infection due to other agents were not excluded? If not excluded, what was the mortality status in this subgroup of cases? Likewise, the total number of SARS-CoV-2 cases is shown as 423 (28%) in Table 1. Why those patients 15 of those excluded from the study included or presented here? To avoid the confounding effect, I advise the authors to reassess all their conclusions regarding these cases.
5. In the discussion section, in line 24, a result is reported as ''COVID-19 vaccination is protective to in-hospital mortality among those with COVID-19 CAP.'' However, in lines 226 and 227, it is stated that being vaccinated against COVID-19, contact with COVID-19 at home or workplace, and the presence of comorbidities such as diabetes mellitus, CKD, malignancy are associated with CAP due to SARS-CoV-2.
The authors must explain why vaccination is both a risk and a preventative measure for CAP due to SARS-CoV-2 in proper sentences in the findings and discussion sections.
6. In addition, as the authors stated, I think that a definite judgment about the vaccine cannot be made with this study due to the different vaccination rates of vaccinated cases and the use of different vaccines.
7. I think that a few language checks need. For example; In terms of the use of The, Have, or Has required a few language checks. For example; The sentence ''The unprecedented situations challenged the healthcare system with a large number of hospitalizations due to CAP because of the unfamiliarity of the disease.'', can be corrected as ''Unprecedented situations have challenged the healthcare system with a large number of hospitalizations due to CAP because of the unfamiliarity of the disease.''
Author Response
21 January 2023
Journal of Clinical Medicine
Dear Reviewer:
I wish to submit the revision of an original research article for publication in Journal of Clinical Medicine titled “Characteristics, outcomes, and factors affecting mortality in hospitalized patients with CAP due to different variants of SARS-CoV-2 and non-COVID-19 CAP.” (revised title). The authors have tried every effort to respond to the suggestion of reviewers and provide the additional data to improve the quality of this manuscript. The responses as follows;
Reviewer 1
Comments and Suggestions for Authors
I evaluated the study.
In this study, the authors present a study called "Features, outcomes, and factors affecting mortality in hospitalized patients with community-acquired pneumonia due to different SARS-CoV-2 variants", as the title suggests.
Some important criticisms of the study, which I have mentioned in the order, are shown below.
- 1. From the name of the study, it is likely understood that all patients consisted of patients with pneumonia caused by COVID-19 variants. However, according to the method part, the rate of CAP associated with COVID-19 is 27%. For this reason, I think it would be more appropriate to change the title. For example; Characteristics, outcomes, and factors affecting mortality in hospitalized patients with CAP due to different variants of SARS-CoV-2 and non-COVID-19 CAP.
Response: Thank you very much for suggestion. We have changed the title as follow; “Characteristics, outcomes, and factors affecting mortality in hospitalized patients with CAP due to different variants of SARS-CoV-2 and non-COVID-19 CAP.”
- 2. Patients excluded from the analysis should not have been mentioned or included anywhere else in the text, except for the method section. However, the authors mentioned 15 cases with co-infection in the results section.
Response: We agree with the reviewer. We have deleted the data of 15 cases with co-infection from the result section.
- 3. As on 130. line stated, If 15 cases had been excluded, it is an erroneous approach to indicate the data of excluded patients in the results section. As I mentioned earlier, the exclusion criteria must only be in the method section. It would be wrong to show excluded cases in the results. This line needs to be corrected. The exclusion criteria should also be reviewed.
Response: We agree with the reviewer. We have added the exclusion in the Method as follows; “The exclusion criteria were patients with 1) < 50% completeness of the data record, 2) an initial diagnosis of hospital-acquired pneumonia or ventilator-associated pneumonia and 3) SARS-CoV-2 co-infection with other pathogens.” In line 77-79.
- 4. In Table 1, All patients (including those with co-infection) =1526-15 (Co-infection) =1511. I don't understand why the authors show the excluded cases along with all cases here in Table 1. Why the patients with co-infection due to other agents were not excluded? If not excluded, what was the mortality status in this subgroup of cases? Likewise, the total number of SARS-CoV-2 cases is shown as 423 (28%) in Table 1. Why those patients 15 of those excluded from the study included or presented here? To avoid the confounding effect, I advise the authors to reassess all their conclusions regarding these cases.
Response: We agree with the reviewer. We have revised Table 1 and Table 2 by excluding the 15 cases with co-infection. We also did not put the data on those patients with co-infection in any analyses of this study to avoid the confounding effect as the suggestion.
- 5. In the discussion section, in line 24, a result is reported as ''COVID-19 vaccination is protective to in-hospital mortality among those with COVID-19 CAP.'' However, in lines 226 and 227, it is stated that being vaccinated against COVID-19, contact with COVID-19 at home or workplace, and the presence of comorbidities such as diabetes mellitus, CKD, malignancy are associated with CAP due to SARS-CoV-2.
The authors must explain why vaccination is both a risk and a preventative measure for CAP due to SARS-CoV-2 in proper sentences in the findings and discussion sections.
Response: We are sorry for unclear communication. We have revised the sentence as follow; “The factors associated with CAP due to SARS-CoV-2 included household or workplace contact with COVID-19, presence of comorbidities such as diabetes mellitus, CKD, and malignancy.” In line 224-226.
- 6. In addition, as the authors stated, I think that a definite judgment about the vaccine cannot be made with this study due to the different vaccination rates of vaccinated cases and the use of different vaccines.
Response: We strongly agree with the reviewer. We have added the sentence in conclusion part as follows; “Regarding to the wide range of vaccine regimens and dosages, even this study demonstrated that the COVID-19 vaccination trended to be protective factor, the efficacy of vaccination should be explored. “in line 411-413. We also revised the abstract as follow; “COVID-19 had the great impact on epidemiology and outcomes of CAP.” in last sentence in abstract.
- 7. I think that a few language checks need. For example; In terms of the use of The, Have, or Has required a few language checks. For example; The sentence ''The unprecedented situations challenged the healthcare system with a large number of hospitalizations due to CAP because of the unfamiliarity of the disease.'', can be corrected as ''Unprecedented situations have challenged the healthcare system with a large number of hospitalizations due to CAP because of the unfamiliarity of the disease.''
Response: Thank you for the suggestion. We have revised the sentences as follows; “Unprecedented situations have challenged the healthcare system with a large number of hospitalizations due to CAP because of the unfamiliarity of the disease.20” in line 51-53. We also sent the manuscript for language check by Editage with certification JOB CODE SNGKL_213.
Sincerely yours,
Sarunyou Chusri, M.D., Ph.D.
Corresponding author

Reviewer 2 Report
This prospective observational study performed in Thailand during a total of 1 year and 6 months( including a large number of patients) aimed to compare the clinical and non-clinical outcomes of hospitalized patients with CAP due to SARS-CoV-2 variants and those with CAP due to other causes. The original also provide a very interesting information, among others, about epidemiology of hospitalized CAP, the rate of coinfections, prognosis factors and the positive impact of COVID vaccination. The manuscript is well designed and structured and methods and results are clearly presented. For these reasons we think the original deserves to be published in this journal. However we would like to make the following minor comments and suggestions in order to improve the quality of the manuscript:
1. One of the most clearly factor associated to mortality is admission in ICU. May you provide us any information about this issue?. Specially focused on number of patients admitted and a subanalysis of these patients based on the rates of mechanical ventilation, incidence of mutiorgan failure, renal failure and need of vassopresors.
2. Treatment with corticosteroids, remdesivir and IL-6 inhibitors had no influence on mortality. Could you give some explanation in the text (discussion)
3. The rate of CAPA is very low comparing with other published papers. Please make a comment in the text
4. Do you have any information about multiresistance as a cause of mortality?
5. The rate of COVID Vaccination is very low. After having your possitive results, we miss a strong recommendation of it in conclusions paragraph.
Author Response
21 January 2023
Journal of Clinical Medicine
Dear Reviewer:
I wish to submit the revision of an original research article for publication in Journal of Clinical Medicine titled “Characteristics, outcomes, and factors affecting mortality in hospitalized patients with CAP due to different variants of SARS-CoV-2 and non-COVID-19 CAP.” (revised title). The authors have tried every effort to respond to the suggestion of reviewers and provide the additional data to improve the quality of this manuscript. The responses as follows;
Reviewer 2
Comments and Suggestions for Authors
This prospective observational study performed in Thailand during a total of 1 year and 6 months( including a large number of patients) aimed to compare the clinical and non-clinical outcomes of hospitalized patients with CAP due to SARS-CoV-2 variants and those with CAP due to other causes. The original also provide a very interesting information, among others, about epidemiology of hospitalized CAP, the rate of coinfections, prognosis factors and the positive impact of COVID vaccination. The manuscript is well designed and structured and methods and results are clearly presented. For these reasons we think the original deserves to be published in this journal. However we would like to make the following minor comments and suggestions in order to improve the quality of the manuscript:
- One of the most clearly factor associated to mortality is admission in ICU. May you provide us any information about this issue?. Specially focused on number of patients admitted and a subanalysis of these patients based on the rates of mechanical ventilation, incidence of mutiorgan failure, renal failure and need of vassopresors.
Response: Thank you very much for the suggestion. We have provided the data on initial ICU admission, mechanical ventilator/ respiratory failure, multi-organ failure and need of vasopressor in Table 1 in the part of clinical data. We have also provided the data on length of ICU stay in Table 4.
- Treatment with corticosteroids, remdesivir and IL-6 inhibitors had no influence on mortality. Could you give some explanation in the text (discussion)?
Response: Thank you very much for suggestion. We have added the discussion for this point as follows; “Although previous studies demonstrated the efficacy of systemic corticosteroids, remdesivir, interleukin-6 inhibitors, and Janus kinase inhibitors for patients with se-vere CAP due to SARS-CoV-281–85, those efficacies were not demonstrated in this current study. It can be explained as follows: 1) due to the wide range of severity of patients in this study, patients with mild disease might not get benefit from those treatments. 2) Most of the patients were treated according to the standard guidelines, which recom-mend using these treatments based on the severity of the patient.” in part of discussion on line 393-399.
- The rate of CAPA is very low comparing with other published papers. Please make a comment in the text
Response: Thank you very much for suggestion. We have added the discussion for this point as follows; “When compared to other earlier studies, this one found a relatively low incidence of COVID-19-associated pulmonary aspergillosis (CAPA), with an estimated incidence of 8.6% in patients receiving mechanical ventilation80. This could be explained by the fact that COVID-19 CAP patients were admitted in a newly constructed isolated or negative pressure room instead of an old one; before COVID-19 pandermic, the hospital had a very low number of negative pressure or isolated rooms. So the new one may have less fungal contamination and reduce the incidence of aspergillosis and mucormycosis.” in part of discussion on line 385-392.
- Do you have any information about multiresistance as a cause of mortality?
Response: We have provided the data on antimicrobial resistance of the organisms causing mortality in Table 5.
- The rate of COVID Vaccination is very low. After having your possitive results, we miss a strong recommendation of it in conclusions paragraph.
Response: We strongly agree with the reviewer. We have added the sentence in conclusion part as follows; “Regarding to the wide range of vaccine regimens and dosages, even this study demonstrated that the COVID-19 vaccination trended to be protective factor, the efficacy of vaccination should be explored. “in line 411-413
Sincerely yours,
Sarunyou Chusri, M.D., Ph.D.
Corresponding author

Round 2
Reviewer 1 Report
Dear Editor
I've seen the authors' replies. They state that they have made the necessary corrections. I reviewed the revised version of the draft. All the criticisms I put forward in the previous observation have been answered. I think that the study should be published in your journal in its current form.
Best wishes.